Holistic Environmental Approaches and Aichi Biodiversity Targets: accomplishments and perspectives for marine ecosystems

Dreujou Elliot elliot.dreujou@uqar.ca 1 2
Carrier-Belleau Charlotte 2
Goldsmit Jesica 2 3
Fiorentino Dario 4 5
Ben-Hamadou Radhouane 6
Muelbert Jose H. 7 8
Godbold Jasmin A. 9
Daigle Rémi M. 2
Beauchesne David 1
1 Institut des Sciences de la Mer, University of Québec at Rimouski , Rimouski , Québec , Canada
2 Department of Biology, Laval University , Québec , Québec , Canada
3 Maurice Lamontagne Institute, Fisheries and Oceans Canada , Mont-Joli , Québec , Canada
4 Helmholtz Institute for Functional Marine Biodiversity, University of Oldenburg , Oldenburg , Germany
5 Alfred Wagner Institute, Helmholtz Centre for Polar and Marine Research , Bremerhaven , Germany
6 Department of Biological and Environmental Sciences, College of Arts and Sciences, Qatar University , Doha , Qatar
7 Instituto de Oceanografia, Universidade Federal do Rio Grande , Rio Grande , Brazil
8 Institute for Marine and Antarctic Sciences, University of Tasmania , Hobart , Australia
9 School of Ocean and Earth Science, University of Southampton, National Oceanography Center , Southampton , United Kingdom
Reimer James
Electronic publication date: 2020 Feb 25
Publication date: 2020
Volume: 8
Electronic Location ID: e8171
Received 2019 Feb 13; Accepted 2019 Nov 6
Copyright: ©2020 Dreujou et al.
Copyright year: 2020
Copyright holder: Dreujou et al.
License: This is an open access article distributed under the terms of the Creative Commons Attribution License, which permits unrestricted use, distribution, reproduction and adaptation in any medium and for any purpose provided that it is properly attributed. For attribution, the original author(s), title, publication source (PeerJ) and either DOI or URL of the article must be cited.
License URL: https://creativecommons.org/licenses/by/4.0/

Keywords: Marine conservation, Research priorities, Holistic approaches, Strategic plan for biodiversity, Aichi Biodiversity Targets

Funding: NSERC Canadian Healthy Oceans Network (CHONe) This work was supported by the 4th World Conference on Marine Biodiversity, and by the NSERC Canadian Healthy Oceans Network (CHONe) and its Partners: Department of Fisheries and Oceans Canada and INREST (representing the Port of Sept-Îles and City of Sept-Îles). The funders had no role in study design, data collection and analysis, decision to publish, or preparation of the manuscript.

==============================
In order to help safeguard biodiversity from global changes, the Conference of the Parties developed a Strategic Plan for Biodiversity for the period 2011–2020 that included a list of twenty specific objectives known as the Aichi Biodiversity Targets. With the end of that timeframe in sight, and despite major advancements in biodiversity conservation, evidence suggests that the majority of the Targets are unlikely to be met. This article is part of a series of perspective pieces from the 4th World Conference on Marine Biodiversity (May 2018, Montréal, Canada) to identify next steps towards successful biodiversity conservation in marine environments. We specifically reviewed holistic environmental assessment studies (HEA) and their contribution to reaching the Targets. Our analysis was based on multiple environmental approaches which can be considered as holistic, and we discuss how HEA can contribute to the Aichi Biodiversity Targets in the near future. We found that only a few HEA articles considered a specific Biodiversity Target in their research, and that Target 11, which focuses on marine protected areas, was the most commonly cited. We propose five research priorities to enhance HEA for marine biodiversity conservation beyond 2020: (i) expand the use of holistic approaches in environmental assessments, (ii) standardize HEA vocabulary, (iii) enhance data collection, sharing and management, (iv) consider ecosystem spatio-temporal variability and (v) integrate ecosystem services in HEA. The consideration of these priorities will promote the value of HEA and will benefit the Strategic Plan for Biodiversity.

Introduction

In 2010, the 10th Conference of the Parties revised and updated the Strategic Plan for Biodiversity from the Convention on Biological Diversity (CBD), which included the Aichi Biodiversity Targets for 2011–2020 (Secretariat of the CBD, 2010). The mission of the Strategic Plan for Biodiversity is to “take effective and urgent action to halt the loss of biodiversity in order to ensure that by 2020 ecosystems are resilient and continue to provide essential services […]” (Secretariat of the CBD, 2010). According to the United Nations (1992), biodiversity refers to “the variability among living organisms from all sources including, inter alia, terrestrial, marine and other aquatic ecosystems and the ecological complexes of which they are part; this includes diversity within species, between species and of ecosystems”. Yet, despite recent small- and large-scale conservation and management efforts, including the development of global protected area networks (Butchart et al., 2015), evidence suggests that most of the Targets are unlikely to be met (Secretariat of the CBD, 2014) as species declines and extinctions continue to occur (Tittensor et al., 2014).

With the end of the Strategic Plan for Biodiversity in sight, the time is ripe to reflect on accomplishments thus far and to identify the next steps towards successful biodiversity conservation in marine ecosystems. These steps will be critical to meet the Sustainable Development Goal 14, which aims for the conservation and sustainable use of the oceans, seas, and marine resources by 2030 (SDG, 2019). These topics were tackled during the 4th World Conference on Marine Biodiversity held in Montréal, Canada, in May 2018, which gathered marine biodiversity experts from around the world. A mentoring program was devised to bring senior and early-career scientists together to address this challenge, which resulted in a series of perspective pieces, including this article. Holistic Environmental Approaches (HEA) were identified as crucial to marine biodiversity conservation by program participants.

In the present study, we define HEAs as environmental planning, assessment, management, or monitoring strategies that use a whole-system approach to explicitly consider and prioritize ecosystem complexity. Holism is dependent on components, connections, and boundaries of the considered ecosystems. While HEAs focus on natural ecosystems, they may include additional dimensions (e.g., social, cultural and economic) relevant to the ecosystem under consideration. There is little doubt that the complexity of ecosystems must be considered for successful marine biodiversity conservation, yet the contribution of HEAs to marine biodiversity conservation in general and to the Aichi Biodiversity Targets in particular is unclear. In this perspective paper, we review the prevalence of HEAs in the peer-reviewed marine biodiversity literature and discuss their relevance to reaching the Aichi Biodiversity Targets, with a focus on the ecological dimension of HEAs. We then propose research priorities to enhance HEAs for marine biodiversity conservation beyond 2020.

Literature Review

Methodology

To better understand the uses of HEAs and their relevance for the Strategic Plan for Biodiversity, we searched the peer-reviewed scientific literature between January 1990 and July 2019 (inclusive). We used the ISI Web of Knowledge database and we queried the title, keywords, and abstract of original research articles. Non-peer reviewed literature, such as technical reports or assessment tools, were not included in this review as we considered that it could produce an important bias by the selection of studies related only to a specific region or for a specific use. We constrained our search to environmental studies by using the search terms ecology, ecosystem, environment, habitat, species and biodiversity as an initial filtering criteria (Table 1). We then further selected articles that focused on marine environments only.

A list of HEAs was established by gathering expert opinion from researchers in the field of marine ecology and environmental conservation. This process led to the inclusion of nine HEAs: adaptive management (Stankey, Clark & Bormann, 2005), cumulative impact assessment (Jones, 2016), ecosystem-based management (Link, 2002; Pikitch et al., 2004; Levin et al., 2009), integrated management (Cicin-Sain & Belfiore, 2005), marine spatial planning (Santos et al., 2019), social-ecological networks (Baggio & Hillis, 2018), strategic environmental assessment (Gunn & Noble, 2009; Gunn & Noble, 2011), sustainable resource management (Bringezu & Bleischwitz, 2009), and systematic conservation planning (Margules & Pressey, 2000). We used all HEA collectively as a search query on the initial corpus, then each HEA was queried individually to determine their prevalence in the literature (Table 1). Finally, we used the search term Aichi targets in order to determine if and how Aichi Biodiversity Targets were considered in HEA studies.

Table 1 Search terms used in the ISI Web of Knowledge to characterize the relevance of Holistic Environmental Approaches (HEAs) to achieving the Strategic Plan for Biodiversity.

The different queries were limited from January 1990 to July 2019 (inclusive). Queries and search terms have been formatted with a regular expression syntax (REGEX) structured with conditional statements in italics, except for queries 2.x which have searched only for one type of HEA at a time.

ID	Query	Articles	
1	CriteriaAND HEAs	1,648	
2	CriteriaAND HEAsAND “marine”	505	
2.1	Adaptive management	69	
2.2	Cumulative impact assessment	2	
2.3	Ecosystem-based management	223	
2.4	Integrated management	43	
2.5	Marine spatial planning	159	
2.6	Social-ecological network	1	
2.7	Strategic environmental assessment	5	
2.8	Sustainable resource management	5	
2.9	Systematic conservation planning	83	
3	CriteriaAND HEAsAND “marine” AND “Aichi”	12	
Notes.

Criteria: (ecolog* OR ecosystem OR environment* OR habitat OR species) AND “biodiversity” carriage return.

HEAs: “adaptive management” OR (“cumulative effect* assessment” OR “cumulative impact* assessment”) OR “ecosystem.based management” OR (“integrated management” OR “integrative management”) OR “marine spatial planning” OR “social.ecological network*” OR “strategic environmental assessment” OR “sustainable resource management” OR “systematic conservation planning”.

Prevalence of HEAs in the marine biodiversity literature

Our review identified 1,648 research articles related to biodiversity studies that used any of the identified HEAs, with 505 articles targeting marine environments. We found that the term ecosystem-based management was the most represented HEA (40.2%), followed by marine spatial planning (31.5%) (Fig. 1). Other HEAs were less represented in the scientific literature, with systematic conservation planning, adaptive management and integrative management referred to in 16.4%, 13.7%, and 8.5% of the identified literature, respectively (Fig. 1). Overall, few studies have considered multiple HEAs simultaneously, with 39 articles having the highest overlap between ecosystem-based management and marine spatial planning. When analyzing the keywords that were used in the reviewed articles, the most prevalent HEAs were “ecosystem-based management” and “marine spatial planning”. Another common keyword was “marine protected areas”, highlighting the relatively common use of this tool in marine conservation programs.

Figure 1 Number of articles per year adopting a Holistic Environmental Approach (HEA) identified in ISI Web of Knowledge.

(A) Number of HEA studies conducted in terrestrial, freshwater and marine environments (light grey), including studies focusing only on marine environments (dark grey). (B) Prevalence of each HEA within studies targeting marine environments only. Searches queried the title, abstract and keywords of peer-reviewed articles. Publication of the Aichi Biodiversity Targets in 2010 is represented by the black dashed vertical line.

The results show that HEAs were rarely discussed before 2006, and the number of HEA articles peaked in 2013, 2014, and 2018 (Fig. 1). Overall, there was a steady increase in the number of HEA articles since 2000. This is particularly true for marine HEAs where the number of studies increased notably two years after the development of the Aichi Biodiversity Targets in 2010 (Fig. 1). This increase after 2012 appears to be largely driven by a rise in the number of ecosystem-based management and marine spatial planning studies, which is likely a reflection of the time required for Aichi-related frameworks to be implemented in research supporting the management of socio-ecological systems (e.g., White et al., 2010).

Of all the studies on HEAs, only 12 specifically used the term Aichi Targets, representing 2.4% of the papers originally identified (Table 2). This is a low proportion of HEAs contributing to the Strategic Plan for Biodiversity, even if we acknowledge that a study does not need to focus on a specific Target to allow a contribution. In addition, nine studies explicitly considered Targets in their research objectives (Table 2). The most frequently mentioned Aichi Biodiversity Target was Target 11, which aims for the conservation of 10% of coastal and marine areas by 2020 (Secretariat of the CBD, 2010). This Target is one of the few that specifically identifies quantitative thresholds for protected areas (Harrop, 2011), which supports the development of well specified and measurable objectives and tools such as simple, measurable, accurate, realistic, time-bound indices (SMART). HEAs could use SMART objectives, although there are few examples of their use in this context (Ehler, 2017). Specifying SMART objectives can be a difficult task, but their measurable component can highlight successful accomplishment of expected thresholds (Ehler, 2017; and references therein). Many studies selected in our literature review evaluated progress and developments of marine protected areas (e.g., Amengual & Alvarez-Berastegui, 2018; Jantke et al., 2018; Rees et al., 2018). Target 11 has also been used to evaluate case studies (Diz et al., 2018), and to identify the sustainable use of specific marine protected areas as part of workshops and wider consultations (Johnson et al., 2014; Sarker et al., 2019). Other selected studies considered either a specific Aichi Biodiversity Target, such as Target 12 in Davidson & Dulvy (2017) or Target 19 in Lagabrielle et al. (2014), or multiple Targets, such as Targets 1, 3, 6, and 17 in Cisneros-Montemayor, Singh & Cheung (2018) or Targets 6, 10, 11, and 12 in Davies et al. (2017) (Table 2). Five articles did not use Aichi Targets in their specific objectives, but were included to set the wider context of the article (e.g., Lagabrielle et al., 2014; Yamakita et al., 2015; Davidson & Dulvy, 2017; Davies et al., 2017; Novaczek et al., 2017) (Table 2).

Table 2 Links between articles adopting a Holistic Environmental Approach (HEA) obtained for Query 3 of the literature review and the Aichi Biodiversity Targets.

ID	Article	Type of HEA considered	Targets considered	Targets as objectives?	
1	Amengual & Alvarez-Berastegui (2018)	Marine spatial planning	11	Yes	
2	Cisneros-Montemayor, Singh & Cheung (2018)	Adaptive management	1, 3, 6, 17	Yes	
3	Davidson & Dulvy (2017)	Systematic conservation planning	11, 12	No	
4	Davies et al. (2017)	Systematic conservation planning	6, 10, 11, 12	No	
5	Diz et al. (2018)	Marine spatial planning	11	Yes	
6	Jantke et al. (2018)	Systematic conservation planning	11	Yes	
7	Johnson et al. (2014)	Ecosystem-based management	6, 11	Yes	
8	Lagabrielle et al. (2014)	Marine spatial planning	11, 19	No	
9	Novaczek et al. (2017)	Adaptive management	11	No	
10	Rees et al. (2018)	Marine spatial planning	11	Yes	
11	Sarker et al. (2019)	Integrated management	11	Yes	
12	Yamakita et al. (2015)	Strategic environmental assessment	11	No	

Linking HEAs and the strategic plan for biodiversity

Strategic Goals have been identified by the CBD as the steps necessary to safeguard biodiversity by 2020 (Fig. 2A). These Goals include mainstreaming biodiversity across government and society (Goal A), reducing direct pressures on biodiversity (Goal B), improving the status of biodiversity (Goal C), enhancing benefits from biodiversity and ecosystem services (Goal D) and enhancing implementation of the established measures (Goal E) (Secretariat of the CBD, 2010). Aichi Biodiversity Targets have been set within each Goal, with specific objectives or quantitative thresholds to reach (Fig. 2B). Our literature review gathered a large number of HEA studies where a few referred to Targets in their objectives and methods (Table 1). Collectively, we found that these studies have focussed on eight Targets, with five being specified as objectives of the study (Table 2, Fig. 2C). This provides examples of how HEAs can contribute to the Strategic Plan for Biodiversity while also providing feedback to reach specific Targets (Figs. 2B–2C). We will discuss some examples of these relationships in more detail in the section below.

Figure 2 Conceptual diagram of interactions and relationships between the Strategic Goals (A), the Aichi Biodiversity Targets (B), Holistic Environmental Approaches (C), and the identified research priorities (D).

Targets have been summarized from Secretariat of the CBD (2010), and the letter before their number corresponds to the Goal to which they belong. Solid arrows represent direct relationships between sections, and dashed arrows represent secondary feedback.

Modern sustainable development objectives include minimizing cross-scale human impacts on biodiversity; concurrently, management plans are increasingly integrating social and economic dimensions (IPCC, 2014; Steffen et al., 2015). Thus, by also considering these same dimensions, HEAs explicitly include stakeholder involvement, public consultations or social initiatives, which is in accordance to Target 1. When made available to the public, the use of a whole-system approach within HEAs, in order to embrace ecosystem complexity, can raise awareness about biodiversity (Palerm, 2000; Portman, 2009; Jarvis et al., 2015). Implementation of conservation actions are usually complicated due to the variety of people concerned and the commercial interests of the different stakeholders (Margules & Pressey, 2000), but also because marine settings are particularly challenging, as stakeholders and objectives tend to be less well-defined (Cisneros-Montemayor, Singh & Cheung, 2018). HEAs that take into account the natural variability of ecosystems, such as adaptive management or ecosystem-based management, should include social and political involvement (Stankey, Clark & Bormann, 2005).

HEAs should also favor whole-system approaches to prioritize management actions based on ecosystem services, which relates to human use of environments (Carpenter et al., 2009; Chan & Ruckelshaus, 2010; Kareiva et al., 2011; Queiroz et al., 2015). Cumulative impact assessments, for example, focus on drivers of change and mechanistic pathways of impact in order to prioritize management efforts and take into account ecosystem services and thus, socio-economic dimensions (Brown et al., 2013; Cook, Fletcher & Kelble, 2014; Cisneros-Montemayor, Singh & Cheung, 2018). These approaches can be linked with Target 3′s objectives to decrease negative effects on biodiversity and encourage conservation and sustainable use of biodiversity.

Target 6 states that fish and invertebrate stocks and aquatic plants are managed and harvested sustainably, legally, and applying ecosystem-based approaches, so that overfishing is avoided, recovery plans and measures are in place for all depleted species, fisheries have no significant adverse impacts on threatened species and vulnerable ecosystems, and the impacts of fisheries on stocks, species and ecosystems are within safe ecological limits (Secretariat of the CBD, 2010). HEAs, such as ecosystem-based management, resource management and adaptive management (along with all the processes linked to theses approaches) will provide the tools to a better understanding of the species, stocks, and habitats as well as their interactions in ecosystems (Arkema, Abramson & Dewsbury, 2006). These tools may be applied to a variety of concrete case studies, ranging from the conservation of marine mammals to coral reef protection (Maggs, Mann & Cowley, 2013; Authier et al., 2017), but also to discuss the adequacy and performance of management strategies (Johnson et al., 2014; Cisneros-Montemayor, Singh & Cheung, 2018).

With the aim of improving the status of biodiversity, governments and companies are required to enforce measures to safeguard ecosystems and all components therein (Secretariat of the CBD, 2010). In this context, HEAs can provide tools to accurately predict ecosystem consequences for systems threatened by multiple drivers of change (Nilsson & Dalkmann, 2001). For example, for Target 11 and the conservation of marine and coastal areas, HEAs have a direct contribution by being related and concerned with management, planning, and conservation. HEAs can also be helpful in the identification and assessment of threats by being able to manage the multiple and simultaneous drivers of change and stress.

The implementation of plans and strategies through participatory actions, such as proposed in Target 17, requires the production of concrete tools to manage environmental use. The correct implementation of HEAs can support the development of ecological indices to integrate different ecosystem components in a coherent methodology, since the need for operational tools within management plans has been highlighted (Arkema, Abramson & Dewsbury, 2006; Cisneros-Montemayor, Singh & Cheung, 2018). In fact, these types of assessments are better undertaken when they are done strategically and expressed in a measurable way, e.g., using SMART objectives (Jones, 2016).

Research Priorities

HEAs should integrate all components of the studied ecosystems. However, logistical, technical and monetary considerations may limit the feasibility of such a goal. Nonetheless, ‘partial’ HEAs are often more valuable than specific environmental assessments (Jones, 2016). The complexity and breadth of knowledge needed for ‘full’ HEAs makes them exceedingly difficult to implement, which may likely explain the relatively small number of studies found applying holistic approaches to ecosystem management (Table 1). In order to achieve the goals set by the Strategic Plan for Biodiversity, there is a need to develop management actions beyond 2020 (Secretariat of the CBD, 2014). Discussions to identify the strategic direction for a post-2020 global biodiversity framework are taking place (e.g., IX Trondheim Conference on Biodiversity), and the need for holistic management actions for a sustainable environment has been highlighted.

With this in mind, research priorities for the application of HEAs in marine environments were identified during the 4th World Conference on Marine Biodiversity as part of a mentoring program. Participants worked individually to identify research priorities before the conference, in order to provide a comprehensive list for the conference. This list was then used by participants to collectively curate a list of the top research priorities. This selection was discussed with conference attendees through panel discussions during the conference and comments were used to refine priorities post-conference. This process yielded a list of five research priorities (Fig. 2D). The steps to undertake, in order to develop and promote the use of holistic approaches for marine biodiversity conservation, are discussed below.

Priority I: expand the use of holistic approaches in environmental assessments

Marine biodiversity spans different levels of biological organization (Hagen et al., 2012). The various biological components of a given ecosystem are continuously interacting with their environment within complex ecological networks. However, many environmental assessments focus on a single species or a single component of the ecosystem, overlooking important abiotic and biotic interactions that significantly affect the way organisms interact with their environment and mediate ecosystem functioning (Crain, Kroeker & Halpern, 2008; Bulleri, 2009; Van der Plas, 2019). Therefore, accurately assessing ecological functioning of marine ecosystems and their environmental, social and economic sustainability requires a holistic approach (Burton et al., 2014; Ma et al., 2017).

Characterization of marine biodiversity and ecosystem functioning can be achieved through theoretical, numerical, experimental or monitoring approaches (Costello et al., 2017; Eriksen et al., 2018). Emerging environmental DNA techniques consisting of DNA metabarcoding and metagenomics (e.g., Thomsen & Willerslev, 2015) offer potentially powerful new tools to monitor marine biodiversity and detect new species introductions. This allows reduced investment in traditional taxonomic techniques and biodiversity sampling and provides new opportunities to assess challenging and remote locations (Brown et al., 2016; Lacoursière-Roussel et al., 2018). Moreover, scientific research vessels now deploy vast arrays of equipment and gears simultaneously to answer increasingly complex research questions about whole ecosystems rather than as individual components (e.g., Pesant et al., 2015). These new emerging methodologies and technologies can complement current holistic approaches such as cumulative effects assessments (Halpern et al., 2008; Halpern et al., 2015) or systematic conservation planning (Margules & Pressey, 2000; Ball, Possingham & Watts, 2009; Daigle et al., 2018). Managers increasingly recognize the need to shift towards holistic approaches to generate informed actions more inclusive of the relationships between ecosystem components than those obtained by traditional single-species efforts (e.g., Manley et al., 2004; Beever, 2006).

The use of HEAs is relevant to all Targets within Goals B and C. In particular, expanding the use of holistic approaches could benefit Target 11’s conservation objectives and perspectives by considering the complexity of the ecosystems (Rees et al., 2018).

Priority II: standardize HEAs vocabulary

What is a “driver of change”, and when does it become a “stressor”? What constitutes an “impact”? The need to adopt a common vocabulary is especially important for multidisciplinary approaches in which communication between actors with a variety of backgrounds is often impeded by semantics (Holt et al., 2011). For example, the scientific community frequently uses the expression “cumulative effects assessment”, but the underlying principles are often poorly understood, which may impact the interpretation of these assessments. Along with the definition of a concept, the origins behind the terminology must be explored and the terms standardized prior to their application across disciplines (Judd, Backhaus & Goodsir, 2015).

Analytical frameworks such as DPSIR (Drivers, Pressures, State, Impact, Response) models are useful for HEAs if all of the included elements are well defined and consistent (Kelble et al., 2013). However, Lewison et al. (2016) and Gari et al. (2014) found that despite the widespread application of individual terms across disciplines and projects, there is still no consensus on the definitions of “pressure” and “impact”. These different interpretations decrease the understanding and operationalization of HEAs across scientists, stakeholders, and decision makers (Gari et al., 2014). The strengths of DPSIR frameworks, such as the capacity to describe linkages between human activity and environmental issues, encourage transdisciplinary research and will benefit many disciplines once its components are clarified (Kelble et al., 2013; Lewison et al., 2016).

While vocabulary standardization does not contribute directly to a specific Aichi Biodiversity Target, it will promote the applicability of HEAs by facilitating communication between actors, which could ultimately be advantageous to all Strategic Goals and Aichi Biodiversity Targets.

Priority III: enhance data collection, sharing, and management practices

The application of HEAs is highly dependent on efficient data collection, sharing, and management. However, constructing large datasets for holistic approaches is a challenging endeavour whose complexity is compounded by decentralized digital infrastructure and heterogenous practices (Wilkinson et al., 2016). To this end, we have identified three steps to promote data collection, sharing, and management efficiency for use in HEAs.

Firstly, it is imperative to develop proper mechanisms to incentivize researchers to share their data publicly. In order to accelerate scientific discoveries and optimize research investments, many scientific journals and governmental agencies have initiated strong policies to promote public data archiving (Tenopir et al., 2011; Poisot, Mounce & Gravel, 2013; Roche et al., 2015). Regardless, many researchers remain reluctant to share their data publicly (Tenopir et al., 2011; Hampton et al., 2013; Roche et al., 2014), highlighting the lack of widespread mechanisms to give proper scientific value to data products (Wilkinson et al., 2016).

Secondly, our digital infrastructure should be improved so that data needed for HEAs are easily and openly accessible to all practitioners, scientists, and the public. We recommend adhering and promoting the FAIR Data Principles, which states that data must be Findable, Accessible, Interoperable, and Reusable (Wilkinson et al., 2016; Tanhua et al., 2019). This emerges as a crucial step to foster proper data management practices and to provide quality data and knowledge relevant to HEAs. Open-access data resources such as the Ocean Biodiversity Information System (OBIS, 2019) and the Global Biodiversity Information Facility (GBIF, 2019) exemplify excellent and easily accessible sources that can be used by researchers to share their data.

Finally, we should strive for global standardization of ocean practices. Defining clear standards and protocols will favour compatibility and pave the way towards efficient HEAs by facilitating the aggregation of local and regional datasets into large, holistic datasets. Initiatives that seek such standardization in practices, such as the Essential Ocean Variables from the Global Ocean Observing System (GOOS, 2019) and the Ocean Best Practices repository (OBP, 2019) from the International Oceanographic Data and Information Exchange, should thus be highly promoted.

Addressing these three steps will enhance data and protocol management, along with knowledge transfer and interoperability, which are necessary for efficient and robust HEAs. This will, in turn, facilitate education and outreach, management and conservation actions, evaluation of ecosystem services, data sharing and capacity building, which are the cornerstones of the Strategic Plan for Biodiversity.

Priority IV: consider ecosystem spatio-temporal variability

Ecosystem studies widely recognize the importance of spatial and temporal scales, as they influence ecosystem components (e.g., fauna, flora), and characterize ecological processes (Legendre, 1993; Hagen et al., 2012; Pittman, 2017). Organism-environment interactions occur across a variety of spatio-temporal scales (e.g., Legendre & Gauthier, 2014; Kraan et al., 2015; Yeager et al., 2017; Ryo et al., 2019), but only few HEA studies have acknowledged the need to consider these variations, for example by comparing different seasons or locations (e.g., De la Vega et al., 2018a; De la Vega et al., 2018b). Despite available methodologies to investigate spatio-temporal patterns within ecosystems (e.g., Baselga, 2010; Legendre & Gauthier, 2014), we are unaware of environmental assessments that investigated multiple spatio-temporal structures concurrently in marine environments.

In addition to the organism-environment interactions, spatio-temporal structures can also affect human activities in an economic context. This can be seen with fisheries management, where activities occur across multiple spatio-temporal scales by involving single boat and fleet activities and managed to exploit targeted resources most efficiently (Hilborn, 2007; Watson et al., 2018). For example, tuna fisheries may be three times more profitable if fishing on strong oceanographic fronts (i.e., Lagrangian coherent structures; Watson et al., 2018). This implies that the effects of a physical feature of the water column can trickle through the local food web, ultimately affecting fisheries profitability at the spatial and temporal scale of the physical feature.

Human activities can interact directly and indirectly with a variety of natural drivers, such as shear stress, storms or currents, at different spatio-temporal scales (Van Denderen et al., 2015; Watson et al., 2018). These interactions may trigger biodiversity responses that consequently appear at different levels of organization, influencing both faunal composition and functions that ultimately impact ecosystem functions and services. In order to develop successful conservation actions, HEAs require further understanding of the spatio-temporal structure of ecosystems and the scales of variability of related ecological patterns and processes, in order to adapt to their variability.

With respect to the Aichi Biodiversity Targets, assessing scales of spatio-temporal variability through HEAs will assist in reducing the impacts of human activities on ecosystems and species (Goal B), and to enhance management strategies to improve the status of critical areas (Goal C).

Priority V: integrate ecosystem services

The concept of “ecosystem services” has initiated the creation of a set of principles to be used by researchers and managers to support ecosystem conservation initiatives (De Groot, Wilson & Boumans, 2002; Beaumont et al., 2007). Ecosystem services are the benefits that humans gain from the natural environment (MEA, 2005). They include provisioning (e.g., production of food or raw materials), regulating (e.g., water purification, carbon sequestration), supporting (e.g., soil production, primary production) and cultural services (e.g., aesthetic, recreation) (Beaumont et al., 2007; Fisher, Turner & Morling, 2009; Atkins et al., 2011; Balmford et al., 2011). Such services may be used to find compromises between providing a hospitable environment for human populations, maintaining ecosystem patterns, and processes within a sustainable range of variation (Beaumont et al., 2007; Cardinale et al., 2012; Norris, 2012). Because ecosystem services consider multiple aspects of the ecosystems within integrative frameworks, they will be highly relevant in HEAs.

Management and consideration of each ecosystem service category is often not equivalent within policy, resulting in a possible mismatch with environmental assessment in terms of spatio-temporal scales (Srivastava & Vellend, 2005; Cardinale et al., 2012). In order to use ecosystem services for biodiversity and ecosystem conservation, many ongoing discussions between stakeholders are seeking a common ground in their respective objectives and agendas (Seddon et al., 2016; Dee et al., 2017a). For example, Holt et al. (2011) quantified the types of services most valued by the local community and stakeholders in a coastal wetland and established the legislative mismatches that exist for protecting those ecosystem processes and functions that are necessary to support the valued benefits. This represents an important step towards integration of ecosystem services in frameworks like HEAs. While we acknowledge the complexity of these discussions and the ongoing research on the topic (e.g., Paterson et al., 2011; Langhans et al., 2019), we emphasize that the integration of ecosystem services by stakeholders and within HEAs will provide a great tool for the Strategic Plan for Biodiversity. To this end, approaches considering ecosystems through network theory may be a great tool to consider the complexity of ecosystems with the plurality of human influences and services (Dee et al., 2017b).

Considering ecosystem services in HEAs will benefit the safeguarding of ecosystems and the maximization of benefits as stated in Goal D. The literature review detected an absence of HEA studies specifically including Targets of this Goal, which highlights the need to better link HEAs and ecosystem services.

Conclusion

Holistic environmental assessments have the potential to enhance marine conservation and management initiatives significantly beyond 2020. The use of HEAs has been increasing steadily over the past decade and is likely related to the establishment of the Strategic Plan for Biodiversity. To date, only a few studies refer to specific Aichi Biodiversity Targets in their research objectives. If included, HEAs could improve ecological research related to these Targets in a variety of ways: from the development of ecological indices and increased understanding of species-ecosystem interaction, to the provision of tools for the prediction of multiple drivers of change and helping the establishment of frameworks for citizen science. All these actions could simultaneously increase understanding of ecosystem complexity in management schemes and decision-making in order to achieve biodiversity goals.

We proposed five research priorities that could increase the effectiveness of HEAs in attaining the Aichi Biodiversity Targets, with respect to their current state of completion. Holistic approaches must appropriately assess the ecological functioning of marine ecosystems and their environmental, social and sustainable economic development. There is a need to standardize the vocabulary used for environmental assessments. Data collection needs to integrate system complexity and data management needs to follow recognized international standards. Marine biodiversity monitoring must consider single and multiple ecosystem components, must observe variability at different scales and should link biodiversity conservation to ecosystem services to support their sustainable uses.

Considering these priorities will help raise the value of HEAs to managers, ensuring greater accuracy and predictive power in environmental management, and could greatly help preparation of the work beyond the Strategic Plan for Biodiversity.

This work is the result of the 4th World Conference on Marine Biodiversity mentoring program. We thank Stephanie Allen, Karen Mooney, Lidia Lins Pereira and Hashim Said, who were involved in the initial discussions from which resulted in this work. We also wish to thank Peter Duinker and Natalie Ban for their help during the literature review, along with two anonymous reviewers for their helpful comments during earlier stages of this manuscript. Finally, we particularly thank Philippe Archambault, Anna Metaxas and Paul Snelgrove for their implication in the mentoring committee, who initiated this project, and their helpful comments and suggestions during the writing process.

Additional Information and Declarations

Competing Interests

Author Contributions

Data Availability

The authors declare there are no competing interests.

Elliot Dreujou conceived and designed the experiments, performed the experiments, analyzed the data, contributed reagents/materials/analysis tools, prepared figures and/or tables, authored or reviewed drafts of the paper, approved the final draft.

Charlotte Carrier-Belleau performed the experiments, analyzed the data, prepared figures and/or tables, authored or reviewed drafts of the paper, approved the final draft.

Jesica Goldsmit and David Beauchesne conceived and designed the experiments, performed the experiments, analyzed the data, prepared figures and/or tables, authored or reviewed drafts of the paper, approved the final draft.

Dario Fiorentino, Radhouane Ben-Hamadou, Jose H. Muelbert and Jasmin A. Godbold performed the experiments, authored or reviewed drafts of the paper, approved the final draft.

Rémi M. Daigle conceived and designed the experiments, performed the experiments, authored or reviewed drafts of the paper, approved the final draft.

The following information was supplied regarding data availability:

No raw data was used; this is a literature review.

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

White et al. (2010) White P Godbold J Solan M Wiegand J Holt A 2010 Ecosystem services and policy: a review of coastal wetland ecosystem services and an efficiency-based framework for implementing the ecosystem approach Ecosystem services: issues in environmental science and technology Harrison RM Hester RE Royal Society of Chemistry United Kingdom 29 51
Wilkinson et al. (2016) Wilkinson MD Dumontier M Aalbersberg IJ Appleton G Axton M Baak A Blomberg N Boiten J-W Da Santos L Bourne PE Bouwman J Brookes AJ Clark T Crosas M Dillo I Dumon O Edmunds S Evelo CT Finkers R Gonzalez-Beltran A Gray A Groth P Goble C Grethe JS Heringa J Hoen P Hooft R Kuhn T Kok R Kok J Lusher SJ Martone ME Mons A Packer AL Persson B Rocca-Serra P Roos M Van Schaik R Sansone S-A Schultes E Sengstag T Slater T Strawn G Swertz MA Thompson M Van der Lei J Van Mulligen E Velterop J Waagmeester A Wittenburg P Wolstencroft K Zhao J Mons B 2016 The FAIR guiding principles for scientific data management and stewardship Scientific Data 3 Article 160018 10.1038/sdata.2016.18 PMC4792175 26978244
Yamakita et al. (2015) Yamakita T Yamamoto H Nakaoka M Yamano H Fujikura K Hidaka K Hirota Y Ichikawa T Kakehi S Kameda T Kitajima S Kogure K Komatsu T Kumagai NH Miyamoto H Miyashita K Morimoto H Nakajima R Nishida S Nishiuchi K Sakamoto S Sano M Sudo K Sugisaki H Tadokoro K Tanaka K Jintsu-Uchifune Y Watanabe K Watanabe H Yara Y Yotsukura N Shirayama Y 2015 Identification of important marine areas around the Japanese Archipelago: establishment of a protocol for evaluating a broad area using ecologically and biologically significant areas selection criteria Marine Policy 51 136 147 10.1016/j.marpol.2014.07.009
Yeager et al. (2017) Yeager LA Cith M McPherson JM Williams ID Baum JK 2017 Scale dependence of environmental controls on the functional diversity of coral reef fish communities Global Ecology and Biogeography 26 1177 1189 10.1111/geb.12628