# Peer review of "Holistic Environmental Approaches and Aichi Biodiversity Targets: accomplishments and perspectives for marine ecosystems"

_PeerJ, doi:10.7717/peerj.8171_

## Round 0.1 · original submission · Major Revisions

I have heard back from two reviewers. One recommends rejection, the other major revisions. However, a reading through their comments find some common points, namely the need for more in-depth and detailed writing and analyses. In particular, I agree with Reviewer 1's comment on measurable targets, and Reviewer 2's idea for a figure to aid readers. Please consider all of their constructive comments carefully. I look forward to seeing a revised version of your work.

· Appeal

Appeal

I thank you for the precisions you sent us about the context of our manuscript and how it can be included in the editorial guidelines of PeerJ. The present letter aims to describe the different actions we propose to take in order to enhance the manuscript and prepare a possible submission Appeal after the reviewers’ comments.

Beforehand, we would like to thank the editor and the two reviewers for their advice and comments, which will definitely improve the quality of our manuscript. After discussions with the other authors, we have highlighted several possible ways to improve the current version of our work, that we present below.

1. We will refine and add definitions of the terms we used, such as “holistic” or “environmental assessment”, as we agree that clarification is required to make the reading and the interpretation easier.

2. Concerning the keyword list, we selected the search string “marine” AND “ecosystem-based”, which yielded the results described in the manuscript. During our work, we considered that holistic environmental assessments (HEA) could fall in the category of ecosystem-based approaches (EBA) studies as they both share important characteristics (such as multi-disciplinarity and the inclusion of different ecosystem components to guide decision makers). However, we acknowledge that different search strings could have been tested, in particular with the keywords “environmental assessment” and “monitoring”. We propose to do the review again, with a discussion about search strings used and a comparison with results obtained using other queries. We have also noted the concern from Reviewer 1 about a possible confusion between “environmental assessments” and “environmental monitoring and evaluation”. We will make sure to clarify our position about this particular issue.

3. Our review was restricted to academic works (e.g. original research papers or literature reviews). Typical reviewing tools do not consider other types of documents (such as technical reports or software) and grey literature systematically, precluding us from properly assessing how HEA are included in this type of literature. We will thus not be able to include these types of documents and tools in the review, but we propose to add a section dedicated to the contribution of grey literature on HEA and EBA to Aichi Targets.

4. This manuscript is a perspective paper initially conducted to present expert opinions on the completion of the Aichi Targets in the context of HEA. We understand that our manuscript needs more discussion on the links between EBA, HEA and Aichi Targets, particularly in the second section of the manuscript. We propose to review this section and discuss the links more in depth. We also propose to present the process used to select research priorities more thoroughly in light of the initial objectives of the WCMB mentoring program.

5. According to Reviewer 2’s comment, we propose to convert Table 1 to a conceptual graph in order to clarify the message we want to convey along with its readability.

6. As a general issue, we will make sure to verify any grammatical issues and refine the phrasing on the entire manuscript.

We are confident that these proposed measures will address any concerns from the reviewers and the editor. We thank you for your time and consideration, and we are looking forward to a positive answer to our Appeal.

Sincerely,

Elliot Dreujou


· · Academic Editor

Reject

Dear authors,

Although your study is very timely and has merits, both reviewers have concluded that it suffers from deficiencies that preclude its publication in the current form. In particular:

* both reviewers indicated that key terms used in this manuscript lack clear and consistent definition. This is crucial as it is not just about terminology but also about interpretation, and it ultimately obscures the scope and objectives of the ms.

* the rationale underlying the keyword list used is not clear enough; both reviewers raised concern about it (e.g. why some terms -a priori very relevant- very not included?) ; the total number of articles included in the review should be made clearer and possibly discussed

* given the topic addressed, it would be needed to better present what key documents and tools are already available to guide HEA and/or EBM studies.

* it is important to more explicitly link the components of HEA to the different Aichi targets, more clearly indicating the causalities
Overall, both reviewers found that parts of the (presentation of) methodology and resulting recommendations are cursory and could benefit from a more in-depth analysis.

I thus have to reject the manuscript, but I think that many reviewers' comments can be useful for you.

Best regards
Xavier

# Reviewer 1 ·

Basic reporting

I have a very hard time to capture the scope and objectives of this paper. Nowhere in the text, “environmental assessments” are explained – but to my knowledge, this refers to “an assessment of the ecological consequences (positive and negative) of a plan, policy, program, or actual (development) projects prior to the decision to move forward with the proposed actions”. However, the only way to let this paper make somehow sense is to swap the wording by “environmental Monitoring and Evaluation”. However, there seems to be additional confusion as – in their methodology/literature review - the authors seem to lump “environmental M&E” with “ecosystem-based approaches”; the latter is actually a management approach. This very essential problem, which is not just about terminology but also about interpretation, undermines the relevance of entire study.

The main hypothesis of this paper seems to be that a more holistic environmental monitoring & evaluation can (directly) lead to better progress towards the Aichi Targets. Intermediate steps – i.e. proposition of management approaches and effectiveness of their implementation, are not discussed. Also, one of the key reasons why targets under the current Strategic Plan for Biodiversity are not met is related to the challenge of developing and using indicators to monitor progress in the targets, and the fact that the current targets are often to vague (not SMART) whereas they should be crisp and measurable (with elements that can be disaggregated) reflecting both processes as well as status outcomes. Again, this has not been considered/discussed.

Furthermore:
- Introduction: I am puzzled about the explanations for the links between the so-called "Environmental Assessments" and each of the Strategic Goals, and targets. I found the causalities rather simplistic and not always clear. An example of a confusing sentence (lines 62-64): “holistic frameworks also help in the development of ecological indices, as they integrate different ecosystem components in a coherent methodology, which is relevant for Target 17”. Also the section on Goals B, C and D (lines 67-75) seems to reflect poor understanding of these strategic goals and underlying targets.
- Research Perspectives: The so-called research priorities identified are not research priorities per se, but just proposals for improved environmental monitoring and evaluation, that are – all together – not innovative.
- The entire paper is also characterized by sloppy (e.g. lines 20-21) and weird phrasing (e.g. lines 109-110; lines 151-153; lines 156-158; 191-194; 289-291).

Experimental design

- Methodology: it is unclear what type of studies were analysed (somewhere it is stated that Ecosystem-based management studies were considered to be equivalent to Holistic Environmental Assessments??). Also, the selection of key words is not explained; the authors themselves determine what should be considered as properly ‘holistic’ (lines 93-94). Robustness of analyses is doubtful.
- The methodology used to define the so-called 'research priorities’ (“gathering input from experts”) is too vague to be meaningful

Validity of the findings

Given major flaws explained under 2. - I don't feel conclusions are sufficiently supported, nor clear.

Reviewer 2 ·

Basic reporting

Please check grammar and tense throughout – e.g. line 28 “continue to provide”, e.g. 292 Management of ecosystem services categories, etc.

Please define or present your definition of environmental assessment in the Introduction, as this term is used in many different ways across different disciplines. Also, your initial definition or description of holistic (e.g line 31-32 or 36-37) should also include reference to the time/temporal component which you refer to later in the discussion.

Would the authors consider creating a conceptual diagram or figure linking the components of HEA to the different Aichi targets? This would send a more powerful and elegant message than just listing the Aichi Targets in Table 1 and then referring to each in reference to HEA in the manuscript text.

Experimental design

Is it not clear why the authors did not include “environmental assessment” in the initial terms search?

Also, in filtering the initial search results for holistic EBM, why not include other similar terms such as “comprehensive”, “multi-” or other relevant term?

Please clarify that in the end, the total number of articles included in the review of fit to the 4 categories of HEA is 77?

Validity of the findings

This manuscript reviewed 77 articles for classification of fit into 4 components of HEAs. The rest of the paper provides 5 recommendations for conducting and improving HEA. However, either before or within each of the resulting 5 recommendations, it would be good to review or present what key documents, tools and datasets are already available to guide HEA and/or EBM studies.

Additional comments

The topic of this paper is important and timely (if not overdue) for assessing global or national progress toward the Aichi Targets. However, the methodology, article classification and resulting recommendations seem cursory and could benefit from a more in-depth analyses. Suggestions include improved definitions, expansion of search terms, a conceptual model linking HEA to the Targets, and a more thorough review of existing tools, data and guidance already available for HEA.

---

## Round 0.2 · Minor Revisions

I have heard back from one of the original reviewers, who finds your work much improved. I agree with their assessment in that the work is much improved, and consider this current version more than half-way to where it needs to get to. The reviewer has provided a few more constructive comments to help you improve the work, and I think a revision should be straightforward enough to accomplish.

Reviewer 2 ·

Basic reporting

The first sentence of the abstract and manuscript refers to Conference of the Parties, but fails to mention under which Convention this CoP is operating (e.g. Convention on Biological Diversity)! This is slightly worrisome!

Line 70 - as this an after the fact perspective, it makes more sense to state that you are discussing the relevance of HEAs for reaching the Targets, rather than their contribution. This should be changed throughout (e.g methodology, etc.).

There are still many grammatical and semantic errors throughout. e.g HEA in the plural should be HEAs.

The last paragraph of the introduction is very confusing. Goals A-E are introduced - followed by HEA examples. However, it is difficult to follow the logic, and as written, the HEA examples do not seem to match up to the Goals. For example Goal E is related to enforcement of government or other conservation measures - but the corresponding HEA example is for predicting ecological consequences of mutliple drivers of change?

Experimental design

The methodology section is clearer and improved with the inclusion of Table 1. However the nested search pattern used is not very clear in the methodolgy text. Figure 2 is a bit confusing - in the legend "each HEA" should be "each of the 9 types of HEA" for example. Do not include methods in the Figure Legend.

The study design is essentially just a key word search. No attempt is made to define the 9 HEA terms or to further analyze or categorize what the resulting articles in the search actually said or did.

Validity of the findings

Figure 2 is interesting and a good start. However, the section labels (A-D) in the legend do not appear correct. Also the stated linkages (e.g. arrows) between Aichi Goals/Targets and HEA types/Research Priorities are not intiutive or clear. For example, how do any of listed Aichi Targets prioritize research in HEA vocabulary?

Discussion of the linkages among the 4 components of figure 2 needs to be better discussed in the text. At the moment, they are each referred to individually at convenient places in the paper, but the figure as a whole and the linkages it shows are never really discussed.

If you could find a way to strengthen these linkages (or redefine them) and give them a more thorough discussion in the text, your conclusions would have more validity, and the overall message of the paper would have more rigour, beyond a key word search.

Additional comments

This revision is much improved, but the linkages among the Aichi Goals/Targets and HEA types/ research priorities are still very weak as presented, and specifically need further thought and discussion. Additionally, there are still many grammatical and semantic errors throughout.

---

## Round 0.3 · accepted · Accept

I have gone over your work, and find it well revised, and am happy to move this into production. Congratulations!